# Continued Vaccine Breakthrough Cases of Serotype 3 Complicated Pneumonia in Vaccinated Children, Portugal (2016–2019)

Catarina Silva-Costa,[a] Joana Gomes-Silva,[a] Marcos D. Pinho,[a] Ana Friães,[a] Mário Ramirez,[a] José Melo-Cristino,[a] on behalf of the Portuguese Group for the Study of Streptococcal Infections and the Portuguese Study Group of Invasive Pneumococcal Disease of the Pediatric Infectious Disease Society

[a]Instituto de Microbiologia, Instituto de Medicina Molecular, Faculdade de Medicina, Universidade de Lisboa, Lisboa, Portugal

Catarina Silva-Costa and Joana Gomes-Silva have contributed equally to this paper. Author order was determined alphabetically.

**ABSTRACT** We previously reported that despite the use of pneumococcal conjugate vaccines (PCVs), vaccine serotypes remained important causes of pneumonia with pleural effusion and empyema (pediatric complicated pneumococcal pneumonia [PCPP]). We cultured and performed PCR on 174 pleural fluid samples recovered from pediatric patients in Portugal from 2016 to 2019 to identify and serotype *Streptococcus pneumoniae*. Most PCPP cases ($n = 87/98$) were identified by PCR only. Serotypes 3 (67%), 14, and 8 (5% each) were the most frequent. Vaccine breakthrough cases were seen among age-appropriately, 13-valent, PCV vaccinated children (median: 3 years, range: 17 months to 7 years), mostly with serotype 3 ($n = 27$) but also with serotypes 14 and 19A ($n = 2$ each). One breakthrough was seen with serotype 14 in an age-appropriately, 10-valent, PCV-vaccinated child and another with serotype 3 in a child to whom the 23-valent polysaccharide vaccine was administered. While the relative risk of serotype 1 PCPP decreased almost 10-fold from the period of 2010 to 2015 to the period of 2016 to 2019 (relative risk [RR] = 0.106), that of serotype 3 PCPP almost doubled (RR = 1.835). Our data highlight the importance of molecular diagnostics in identifying PCPP and document the continued importance of serotype 3 PCPP, even when PCV13 use with almost universal coverage could be expected to reduce exposure to this serotype.

**IMPORTANCE** The use of conjugate vaccines against *Streptococcus pneumoniae* in children has led to substantial reductions in pneumococcal invasive disease. However, the reductions seen in each of the 13 serotypes currently included in the highest-valency vaccine approved for use in children (PCV13), were not the same. It is becoming clear that most vaccine breakthroughs worldwide involve serotype 3 and are frequently associated with complicated pneumonia cases, often with empyema or pleural effusion. Here, we show that despite almost universal PCV13 use, which would be expected to reduce vaccine serotype circulation and further reinforce vaccine direct protection, pneumococci and serotype 3 remain the major causes of pediatric complicated pneumonia. Molecular methods are essential to identify and serotype pneumococci in these cases, which frequently reflect vaccine breakthroughs. A broader use of molecular diagnostics will be essential to determine the role of this important serotype in the context of PCV13 use in different geographic regions.

**KEYWORDS** pneumonia, *Streptococcus pneumoniae*, diagnostics, epidemiology, nucleic acid technology, pediatric infectious disease, serotypes, vaccines

Address correspondence to Mário Ramirez, ramirez@fm.ul.pt.

The authors declare a conflict of interest. J.M.-C. has received research grants administered through his university and received honoraria for serving on the speakers' bureau of Pfizer and Merck Sharp and Dohme. M.R. has received honoraria for serving on the speakers' bureau of Pfizer and Merck Sharp and Dohme and for serving in specialist panels of GlaxoSmithKline and Merck Sharp and Dohme. The funders had no role in the design of the study; in the collection, analyses, or interpretation of data; in the writing of the manuscript, or in the decision to publish the results.

*S*treptococcus pneumoniae (pneumococcus) is a leading cause of pneumonia in children worldwide and the most common pathogen isolated in pleural effusions and empyemas, frequently followed by *Streptococcus pyogenes* and *Staphylococcus aureus* (1–4). For this report,

**TABLE 1** Number of requests for laboratory testing and confirmed positive cases of
*Streptococcus pneumoniae* infection in children (Portugal, 2016 to 2019)

| Cases and requests[a] | Year | | | | Total |
|---|---|---|---|---|---|
| | 2016 | 2017 | 2018 | 2019 | |
| Positive for *S. pneumoniae* | 26 | 22 | 22 | 28 | 98 |
| Negative for *S. pneumoniae* | 14 | 23 | 15 | 24 | 76 |
| Requests | 40 | 45 | 37 | 52 | 174 |

[a]Among these cases are 11 for which pneumococci were cultured from pleural fluid (2016, $n = 4$; 2017, $n = 2$; 2018, $n = 1$; and 2019, $n = 4$).

we will refer to pediatric pneumococcal pneumonia occurring with either parapneumonic effusion or empyema as pediatric complicated pneumococcal pneumonia (PCPP).

Despite most cases of pediatric complicated pneumonia having pneumococcal etiology, only a few of the >100 known pneumococcal serotypes are responsible for most cases: serotypes 1, 3, 7F, 14, and 19A (5), all of which are included in the 13-valent pneumococcal conjugate vaccine (PCV13). Even before the availability of the 7-valent pneumococcal conjugate vaccine (PCV7) the incidence of PCPP was increasing in several countries, and this continued despite the use of PCV7 (1, 6), which did not include most of the serotypes frequently implicated in PCPP (5). With the advent of PCV13, which included all the leading serotypes found in PCPP, it was expected that its use would lead to decreases in PCPP cases and changes in the serotypes responsible for these infections. In the United States, there was indeed a significant decrease in PCPP following the introduction of PCV13, mostly due to a decrease in serotype 19A (7, 8). In Taiwan, while PCV13 was available in the private market there was no significant impact in PCPP; however, following its introduction in the national immunization program (NIP) with a catch-up program for children 2 to 5 years old, there was an immediate sharp decrease in PCPP (3). In Spain a similar decrease of PCPP occurred, mostly due to decreases in serotype 1 (9). In contrast, in Germany, where an initial decline in PCPP was seen following PCV13 use, PCPP cases rebounded in recent years (2). Similarly, in Australia, although there was a decrease in all-cause hospitalizations for bacterial pneumonia following PCV13 introduction, empyema hospitalizations increased (1). Cultures of pleural fluid or blood from PCPP case patients are frequently negative, so serologic or nucleic acid diagnostic methods are essential for identifying PCPP cases and the serotypes involved (1, 4, 6, 9). In countries where serotype information is available, most PCPP cases following PCV13 use were due to serotype 3, even among vaccinated children (1, 4, 6, 9). While in some countries this was due to a stable number of serotype 3 PCPP cases (9), others have seen increases in serotype 3 PCPP cases in recent years (1, 6).

In Portugal, PCV7 became available in 2001, the 10-valent PCV (PCV10) in mid-2009 and PCV13 in early 2010. Despite PCVs being offered in the private market without any reimbursement, vaccination uptake varied between 75% and 61% until 2014 (4). In July 2015, PCV13 was introduced in the NIP for children born after January 2015 in a 2 + 1 schedule, with doses given at 2, 4, and 12 months of age, quickly reaching >95% coverage (10). In 2010 to 2015, before PCV13 was included in the NIP, we found that serotypes 1, 3, and 19A were the dominant causes of PCPP in children (<18 years old) and that several cases were PCV13 vaccine breakthroughs, mostly involving serotype 3 (4). We conducted a prospective surveillance study to evaluate the effects of introducing PCV13 into the NIP on PCPP in Portugal.

## RESULTS

**Samples.** Between January 2016 and December 2019, we analyzed 174 pleural fluid samples, from which 11 PCPP cases were identified by culture. The number of samples received each year and the number of PCPP cases are indicated in Table 1.

Among the 98 PCPP cases, 52 were detected in male patients (53%). The gender of one patient was not available. Patient age ranged from 10 months to 16 years, with a median age of 3 years (interquartile range: 2 to 5 years; the age of 1 patient was not available); 14 cases were diagnosed in children <2 years old (14.3%) and 19 among patients ≥6 years old (19.4%).

**TABLE 2** Serotype distribution among the 98 pediatric case-patients with *Streptococcus pneumoniae* infection included in this study, by PCV vaccination status (Portugal, 2016 to 2019)[a]

| Serotype | Age appropriately | | Not age appropriately, PCV13 | Unclear | | | Not vaccinated | Unknown | Total |
| | PCV13 | PCV10 | | PCV13 | PCV10 | PCV7 | | | |
|---|---|---|---|---|---|---|---|---|---|
| 14 | 2 | 1 | | | | | | 2 | 5 |
| 6B | | | | | | | 1 | | 1 |
| PCV7 | 2 | 1 | | | | | 1 | 2 | 6 |
| 1 | | | | | | | 1 | 1 | 2 |
| PCV10 | | | | | | | 1 | 1 | 2 |
| 3 | 27 | | | 2 | 2 | 1[b] | 11 | 23 | 66 |
| 19A | 2 | | | | | | 1 | | 3 |
| PCV13 | 29 | | | 2 | 2 | 1 | 12 | 23 | 69 |
| 8 | 1 | | 1 | | | | 1 | 2 | 5 |
| 11A/11D | 1 | | | | | | | | 1 |
| 15A/15F | | | | | | | | 1 | 1 |
| 22F/22A | | | | | | | 1 | | 1 |
| 9N | | | | | | | | 1 | 1 |
| NVT | 2 | | 1 | | | | 2 | 4 | 9 |
| Not identified | 5 | | | | | | | 7 | 12 |
| Total | 38 | 1 | 1 | 2 | 2 | 1 | 16 | 37 | 98 |

[a]PCV, pneumococcal conjugate vaccine; PCV7, 7-valent PCV; PCV10, 10-valent PCV; PCV13, 13-valent PCV.
[b]Patient also vaccinated with the 23-valent polysaccharide pneumococcal vaccine.

**Serotyping.** We were able to serologically identify the serotype in all ($n = 11$) isolates available and by real-time PCR (RT-PCR) in 86.2% ($n = 75$) of the samples where *S. pneumoniae* was detected by RT-PCR (Table 2). In the remaining 12 pleural fluid samples, most likely the serotype of the causative pneumococcus was not included in those covered by the RT-PCR schema used and these samples were classified as "not identified." Since our RT-PCR schema includes all PCV13 serotypes these were considered non-vaccine serotypes (NVTs) for further analysis. Among the typeable samples and isolates, serotype 3 was the most prevalent, being responsible for 67.3% ($n = 66$) of cases. Serotypes 14 and 8 ranked second, responsible for 5 PCPP cases each. Only serotypes 3, 8, 14, and 9N were detected among culture-positive samples, and all except 9N, which is not included in the RT-PCR reactions, were also detected among culture-negative samples. Overall, the addPCV13 serotypes (3, 6A, and 19A) were responsible for most infections ($n = 69$, 70.4%). PCV7 serotypes accounted for 6.1% of cases ($n = 6$) and the addPCV10 serotypes (1, 5, 7F) accounted for an even smaller proportion ($n = 2$, 2%). In 21 cases (21.4%), the infection was caused by an NVT, a higher proportion than in 2010 to 2015 (11%) (4), but the difference was not statistically significant ($P = 0.056$).

**Vaccination status.** Vaccination status was unknown in 37 cases (37.8%), with 40% having unknown information occurring in 2019 due to a breakdown in reporting from pediatric departments. However, most cases in 2019 ($n = 13/15$) occurred in children ≤4 years old, which we would anticipate to have been vaccinated given the high (>95%) national vaccination coverage since its introduction in the NIP in 2015. A total of 16 cases (16.3%), occurred in nonvaccinated children (Table 2). Among the nonvaccinated patients, serotype 3 was the most frequent ($n = 11/16$). Most of the vaccinated patients had received PCV13; one patient received 1 dose of PCV7 and one dose of PPV23, and 3 patients had received PCV10. In $n = 27/38$ age-appropriately PCV13-vaccinated patients, infections were caused by serotype 3 and were found in all study years. In 13 cases, the patients had received 4 PCV13 doses, with the remaining ones having received 3 doses, mostly already under the NIP schedule. Other vaccine serotypes found in the age-appropriately PCV13-vaccinated patients included serotypes 19A ($n = 2$, both cases in 2018) and 14 ($n = 2$, cases in 2016 and 2019). The median age of children age-appropriately vaccinated with PCV13 where a PCV13 serotype was identified was 3 years (range: 17 months to 7 years). In patients whose vaccinal status was unclear, serotype 3 was found in the 2 patients who had received 2 doses of PCV13. In the only child not age-appropriately vaccinated with PCV13, serotype 8 was detected.

Only three patients had been immunized with PCV10 (Table 2). In the age-appropriately vaccinated patient, serotype 14 was identified, representing a vaccine breakthrough. The other patients had unclear vaccination status, having received only one dose of PCV10 at an unknown date of administration, and serotype 3 was identified as the cause of PCPP in both.

When analyzing the yearly distribution of cases by age group (see table S1 in Text S1 in the supplemental material), there were no differences in the proportion of PCPP cases in each group throughout the study period. The same analysis performed on the serotype distribution of all patients (<18 years) revealed that for serotype 3, there was no change in 2016 to 2019 (Cochran-Armitage [CA] test: $P = 0.260$) (Table S2, Text S1). On the other hand, the relative risk of PCPP caused by serotype 3 increased almost 2-fold in 2016 to 2019 compared to that in 2010 to 2015 (relative risk [RR] = 1.835; 95% confidence interval [CI] = 1.384 to 2.434), while that of serotype 1 decreased 10-fold (RR = 0.106; 95% CI = 0.026 to 0.440). Besides the disappearance of serotype 7F/7A cases in 2016 to 2019, no other significant changes in RR were seen for any other serotype.

## DISCUSSION

As reported previously in Portugal (4) and in most other countries (1, 6, 9), we found that cultures of pleural fluid or blood in PCPP cases are frequently negative and that non-culture methods are essential to identify the etiology of most PCPP cases. This raises the possibility of greatly underestimating the importance of PCPP in overall pediatric (<18 years old) invasive pneumococcal disease (pIPD) in settings where non-culture methods are not used systematically. Consistent with this possibility, we found that almost a third of pIPD cases in Portugal in 2015 to 2018 were due to PCPP detected using molecular methods (11). Molecular diagnostic methods to identify and serotype PCPP may be particularly important to evaluate the contribution of serotype 3 to pIPD, since it is infrequently identified in culture-positive samples but readily detected by PCR-based assays, even in samples from cases with no prior antimicrobial treatment (12). Illustrating this, serotype 3 was the most frequent cause of pIPD in Portugal in 2015 to 2018, and most cases were detected by molecular methods ($n = 43/59$, 72.8%) (11).

The median age of the children with PCPP in Portugal decreased from 4 years in 2010 to 2015 to 3 years in 2016 to 2019. The serotypes also changed from the previous study (2010 to 2015) (4) to the current period, with the RR of serotype 1 and 7F/7A PCPP decreasing (although the latter not significantly) and that of serotype 3 PCPP increasing. Serotype 1 was historically the dominant serotype in PCPP (5) and the decrease seen here, and also in Spain (9), is consistent with PCV13 protection against infection by this serotype. Serotype 7F (included in the 7F/7A cases reported previously) was also among the leading PCPP serotypes worldwide (5) and its disappearance is similarly consistent with the PCV13 effect. The remaining major serotypes implicated in PCPP (5) —3, 14, and 19A—although included in PCV13, did not show signs of further decreases in 2016 to 2019. Similarly to what we found in Portugal, in Australia and Germany the proportion of PCPP cases caused by serotype 3 has increased in recent years (1, 6). In contrast to what was seen when considering all pIPD in Portugal (11) and elsewhere, serotypes not included in PCV13 did not increase as causes of PCPP, suggesting that none of the emerging serotypes have a particular tropism for the pleural space.

The introduction of PCV13 into the NIP did not lead to further reductions in the number of reported PCPP cases, as might have been expected given that most cases before the increased uptake of PCV13 were due to serotypes included in the vaccine (4), and the decreased circulation of these serotypes due to higher vaccine uptake could impact PCPP. In fact, most of the serotype 3 PCPP cases occurred in age-appropriately vaccinated children who had received either 3 or 4 doses of PCV13, including a booster dose, and would be therefore expected to be maximally protected. The effectiveness of PCV13 against serotype 3 invasive disease has been a matter of debate given the persistence of this serotype as a cause of pIPD in multiple countries (9, 11), but a recent review arrived at a pooled vaccine effectiveness (VE) estimate of 63.5% (95% CI = 37.3 to 89.7%) (13). Despite these encouraging results, if non-culture methods are not used for diagnosis, this could greatly influence the estimates of VE against serotype 3

disease, given the high fraction of this serotype's invasive pneumococcal disease (IPD) which is detected solely by molecular or serologic methods. A greater effort in the use of molecular methods in IPD surveillance could help clarify this important issue.

The protection afforded by conjugate vaccines was never anticipated to be complete, and numerous studies have reported vaccine breakthroughs. In the US, most vaccine breakthrough cases in children <5 years old occurred during the first year of life and were due to serotypes 19F and 6B with PCV7 vaccination, and serotypes 19A and 3 with PCV13 vaccination (14). In Ireland, in patients ≤16 years old, most PCV13 vaccine breakthroughs occurred in cases of bacteremia without a focus by serotype 19A, although cases with serotypes 3, 6B, 7F, 14, and 19F were also found (15). In Australia, in empyema cases in patients ≤19 years old, most PCV13 vaccine breakthroughs were serotype 3 cases, but there were also cases with serotypes 14, 19A, and 19F (1). In Germany, also only in empyema cases in patients <18 years old, most vaccine breakthroughs were serotype 3 cases, with a minority occurring with serotypes 19A and 1 (6). In Spain, in patients aged 2 to 59 months, 24 PCV13 vaccine breakthroughs were detected, most associated with complicated pneumonia and serotypes 3 and 19A, although serotypes 1, 6B, and 14 were also detected (16). Similarly to our previous findings (4), serotype 3 continues to be responsible for most PCPP vaccine breakthroughs ($n = 27/31$) in Portugal. The high proportion of pIPD caused by this serotype (11) and the large number of vaccine breakthroughs documented in fully vaccinated children despite 4-years of PCV13 use in the NIP suggest that no further benefits of vaccination can be expected against this serotype. Perhaps more surprising was the identification of PCV13 vaccine breakthroughs with serotypes 14 and 19A ($n = 2$ each). Although these are historically important serotypes in PCPP (5) these serotypes were responsible for a minority of cases from 2010 to 2015 in Portugal ($n = 3$ and $n = 7$, respectively) and only a single vaccine breakthrough was associated with serotype 14 at that time (4). Both these serotypes were also found in vaccine breakthroughs elsewhere, but frequently serotype 19A had been a dominant serotype in PCPP (7, 8), which was not the case in Portugal, even before the introduction of PCV13. A vaccine breakthrough with serotype 14 was detected in a fully vaccinated PCV10 child and another with serotype 3 in a child immunized with PCV7 and PPV23, the latter expected to protect against serotype 3.

Our study has several limitations. Although it was prospective and involved both pediatric and microbiology departments, it was not designed to estimate the incidence of PCPP, since it did not identify cases in which there were clinical or radiographic criteria for complicated pneumonia or those for which pneumococci were identified in either blood or respiratory samples. However, the stable nature of the surveillance suggests that the increase in the number of serotype 3 cases could reflect an increased incidence of serotype 3 PCPP, as seen elsewhere (1, 6). We did not collect information on the immune status or other comorbidities of age-appropriately vaccinated children with PCPP by vaccine serotypes. However, given the high prevalence of serotype 3 in this group, it is unlikely that all cases could be explained by host characteristics and are consistent with the pneumococcal serotype dynamics together with the properties of serotype 3 being responsible for the increased number of cases and of vaccine breakthroughs.

The resilience of serotype 3 in PCPP may have important consequences on the potential overall benefits of PCV13. Current VE estimates are based mostly on data from traditional culture-based microbiological methods (13). Since it is clear that serotype 3 PCPP would be severely underestimated under these conditions (1, 4, 6, 9, 12), this could potentially lead to overestimation of the VE against this serotype. Considering that vaccine breakthroughs occur with a 3 + 0 "unboosted," but also with 2 + 1 and 3 + 1 boosted schedules, the PCV13 administration calendar seems to have only a minor influence on vaccine breakthroughs (1, 4, 14–16). Taken together, these data suggest that PCR-based methods should be more broadly used to identify the etiology of pediatric complicated pneumonia and of the serotypes which cause PCPP to better evaluate the impact of PCV13 on these important infections.

## MATERIALS AND METHODS

**Patient samples.** We requested the submission of all culture-negative pleural effusion samples for which an infectious etiology was suspected, and all pneumococcal isolates grown from pleural fluid samples of

patients <18 years old from 61 hospital laboratories and pediatric departments across all regions of Portugal. No criteria for sample collection were provided, so these may be different across hospitals. Samples were recovered between January 2016 and December 2019, but no audit was performed to evaluate reporting compliance. However, since our network includes all secondary and tertiary care hospitals in which PCPP is likely to be treated and a pleural fluid sample obtained, we assume that our catchment population is the entire population of Portugal in this age group (<18 years). There were no changes in the surveillance network from our previous study (4).

Vaccination status was obtained from the register by the attending physician. In order to define age-appropriately vaccinated children with PCV7, PCV10, or PCV13, we used the guidelines from the Vaccines Committee of the Portuguese Pediatric Infectious Diseases Society (4, 17). For children born before 2015 who received 2 doses of PCV13 or 1 dose of PCV10 or PCV7, vaccinal status was defined as unclear if the date of vaccination was unknown. Children vaccinated according to the NIP received PCV13 in a 2 + 1 schedule (2, 4 and 12 months) (18) and were considered age-appropriately vaccinated, allowing for a 1-month delay in vaccination for the second and third doses relative to the schedule.

This study was approved by the Institutional Review Board of the Centro Académico de Medicina de Lisboa (246/17).

**Culture-positive samples.** *S. pneumoniae* isolates recovered from pleural fluid samples were identified by colony morphology and hemolysis on blood agar plates, optochin susceptibility, and bile solubility. Serotyping was performed with a standard capsular reaction test using the chessboard system and specific serum samples (Statens Serum Institute, Copenhagen, Denmark). Serotype groups were defined according to their inclusion in currently available PCVs: PCV7 (4, 6B, 9V, 14, 18C, 19F and 23F), PCV10 (the PCV7 serotypes and the addPCV10 serotypes: 1, 5, 7F), and PCV13 (the PCV10 serotypes and the addPCV13 serotypes: 3, 6A, 19A). All other serotypes were considered non-vaccine serotypes.

**Culture-negative samples.** When the identification of the etiology by culture failed, the pleural fluid samples were sent to the central laboratory and tested for the presence of *S. pneumoniae* (4). Positive samples were serotyped by real-time PCR as described previously (4). We performed 7 multiplex reactions targeting 3 serotypes or serogroups each: 3, 7F/7A, and 19A; 1, 15B/15C, and 23F; 14, 18, and 19F; 4, 6, and 9V/9A; 5, 11A/11D, and 16F; 8, 12F/12A/12B/44/46, and 22F/22A; and 15A/15F, 23A, and 33F/33A/37. The RT-PCR scheme for serotyping did not enable discrimination between some serotypes within or across a few serogroups, as indicated in the description of each multiplex reaction.

We defined negative results as those with cycle threshold ($C_T$) of ≥40 and positive results as those with a $C_T$ of ≤35 (4). If a $C_T$ value of >35 and <40 was obtained, we considered the result inconclusive and varied the amount of DNA in the reaction by using twice as much DNA and diluting over a 50-fold range. If still no reaction yielded a $C_T$ of ≤35, we considered the sample negative.

**Statistical methods.** We used the Fisher exact test to evaluate differences in the prevalence of the most frequent serotypes, as well as the number of PCPPs among vaccinated and nonvaccinated children, and the Cochran-Armitage test to evaluate trends. We calculated the relative risks and respective 95% confidence intervals of a PCPP case being caused by a particular serotype in the period of 2016 to 2019 versus the period of 2010 to 2015. We considered a $P$ value of <0.05 to be statistically significant.

## SUPPLEMENTAL MATERIAL

Supplemental material is available online only.
**SUPPLEMENTAL FILE 1**, PDF file, 0.1 MB.

## ACKNOWLEDGMENTS

Partial support was received from the Fundação para a Ciência e a Tecnologia (PTDC/DTP-EPI/1555/2014) and from an investigator-initiated grant from Pfizer.

The Portuguese Group for the Study of Streptococcal Infections consisted of the following: Margarida Pinto, João Marques, Isabel Peres, Teresa Pina, Isabel Lourenço, and Cristina Marcelo (Centro Hospitalar de Lisboa Central, Lisboa, Portugal); Marília Gião and Rui Ferreira (Centro Hospitalar do Algarve, Faro e Portimão, Portugal); Rui Tomé Ribeiro, Celeste Pontes, Luísa Boaventura, Catarina Chaves, Teresa Reis, and Henrique Oliveira (Centro Hospitalar e Universitário de Coimbra, Coimbra, Portugal); Ana Cristina Silva, Hermínia Costa, Maria Fátima Silva, and Amélia Afonso (Centro Hospitalar de Entre Douro e Vouga, Santa Maria da Feira, Portugal); Natália Novais and Isabel Brito (Hospital Distrital da Figueira da Foz, Figueira da Foz, Portugal); Luís Marques Lito and Ana Bruschy Fonseca (Centro Hospitalar Lisboa Norte, Lisboa, Portugal); Maria Ana Pessanha, Elsa Gonçalves, Teresa Morais, and Cristina Toscano (Centro Hospitalar Lisboa Ocidental, Lisboa, Portugal); Paulo Lopes, Angelina Lameirão, Gabriela Abreu, and Aurélia Selaru (Centro Hospitalar de Vila Nova de Gaia/Espinho, Vila Nova de Gaia e Espinho, Portugal); Paula Mota and Margarida Tomaz (Centro Hospitalar do Alto Ave, Guimarães, Portugal); Rosa Bento (Centro Hospitalar do Baixo Alentejo, Beja, Portugal); Maria Helena Ramos and Ana Paula Castro (Centro Hospitalar do Porto, Porto, Portugal); Fernando Fonseca (Centro Hospitalar da Póvoa do Varzim/Vila do Conde, Póvoa do Varzim e Vila do Conde, Portugal); Ana Paula Castro (Centro Hospitalar de

Trás-os-Montes e Alto Douro, Vila Real e Peso da Régua e Chaves, Chaves, Portugal); Nuno Canhoto, Filipa Vicente, and Margarida Pereira (Hospital Central do Funchal, Funchal, Portugal); Ilse Fontes and Paulo Martinho (Hospital de Santa Luzia, Elvas, Portugal); Ana Domingos, Gina Marrão, and José Grossinho (Hospital de Santo André, Leiria, Portugal); Manuela Ribeiro and Helena Gonçalves (Centro Hospitalar de São João, Porto, Portugal); Alberta Faustino and Maria Cármen Iglesias (Hospital de Braga, Braga, Portugal); Maria Paula Pinheiro and Rui Semedo (Hospital Dr. José Maria Grande, Portalegre, Portugal); Adriana Coutinho (Hospital do Espírito Santo, Évora, Portugal); Luísa Gonçalves and Olga Neto (Hospital dos SAMS, Lisboa, Portugal); Luísa Sancho (Hospital Dr. Fernando da Fonseca, Amadora, Portugal); José Diogo and Ana Rodrigues (Hospital Garcia de Orta, Almada, Portugal); Elmano Ramalheira, Raquel Diaz, Sónia Ferreira and Inês Cravo Roxo (Hospital Infante D. Pedro, Aveiro, Portugal); Isabel Vale, Ana Carvalho, and José Miguel Ribeiro (Hospital de São Teotónio, Viseu, Portugal); Maria Antónia Read, Valquíria Alves, and Margarida Monteiro (Hospital Pedro Hispano, Matosinhos, Portugal); Margarida Rodrigues (Hospital Reynaldo dos Santos, Vila Franca de Xira, Portugal); José Mota Freitas and Sandra Vieira (Centro Hospitalar do Alto Minho, Ponte de Lima e Viana do Castelo, Portugal); Elsa Calado, Maria Favila Meneses, and José Germano de Sousa (Hospital CUF Descobertas, Lisboa, Portugal; Laboratórios Germano de Sousa, Portugal); Mariana Viana, Marvin da Silva Oliveira, Vanessa Pereira, and Hugo André Macedo (Centro Hospitalar do Tâmega e Sousa, Amarante e Guilhufe, Portugal); Paulo Paixão, Vitória Rodrigues, Sofia Marques, Joana Selada, and Patrícia Pereira (Hospital Beatriz Ângelo, Loures, Portugal; Hospital de Cascais, Cascais, Portugal; Hospitais Lusíadas, Portugal; Hospitais Luz, Portugal); Jesuína Duarte (Centro Hospitalar de Setúbal, Setúbal, Portugal); Paula Pinto (Hospital Distrital de Santarém, Santarém, Portugal); Ezequiel Moreira (Centro Hospitalar do Médio Ave, Santo Tirso e Vila Nova de Famalicão, Famalicão, Portugal); and Adília Vicente (Centro Hospitalar do Oeste Norte, Caldas da Rainha, Portugal).

The Portuguese Study Group of Invasive Pneumococcal Disease of the Pediatric Infectious Disease Society consisted of the following: Maria J. Brito, Teresa Tomé, and Mónica Rebelo (Centro Hospitalar de Lisboa Central, Lisboa, Portugal); Sónia Aires (Centro Hospitalar de Entre Douro e Vouga, Santa Maria da Feira, Portugal); Cristina Ferreira (Centro Hospitalar do Alto Ave, Guimarães, Portugal); Eurico Gaspar (Centro Hospitalar de Trás-os-Montes e Alto Douro, Vila Real e Peso da Régua e Chaves, Chaves, Portugal); Fernanda Pereira (Centro Hospitalar do Nordeste, Bragança, Macedo de Cavaleiros e Mirandela, Portugal); Maria José Dinis (Centro Hospitalar da Póvoa do Varzim/Vila do Conde, Póvoa do Varzim e Vila do Conde, Portugal); Paulo Teixeira (Centro Hospitalar do Médio Ave, Santo Tirso e Vila Nova de Famalicão, Famalicão, Portugal); José Amorim (Centro Hospitalar do Alto Minho, Ponte de Lima e Viana do Castelo, Portugal); Cláudia Monteiro (Centro Hospitalar do Tâmega e Sousa, Amarante e Guilhufe, Portugal); Isabel Carvalho (Centro Hospitalar de Vila Nova de Gaia/Espinho, Vila Nova de Gaia e Espinho, Portugal); Sofia Arosa (Hospital Pedro Hispano, Matosinhos, Portugal); Álvaro Sousa, Margarida Guedes, Laura Marques, and Ana Braga (Centro Hospitalar do Porto, Porto, Portugal); Margarida Tavares (Centro Hospitalar de São João, Porto, Portugal); Isabel Cunha (Hospital de Braga, Braga, Portugal); Lurdes Vicente (Hospital Amato Lusitano, Castelo Branco, Portugal); Maria Manuel Zarcos (Hospital de Santo André, Leiria, Portugal); Helena Almeida (Centro Hospitalar do Oeste Norte, Caldas da Rainha, Portugal); Silvia Almeida (Hospital Infante D. Pedro, Aveiro, Portugal); Eulália Afonso, Fernanda Rodrigues and Cristina Resende (Centro Hospitalar e Universitário de Coimbra, Coimbra, Portugal); Luísa Mendes (Hospital Distrital da Figueira da Foz, Figueira da Foz, Portugal); Cristina Faria (Hospital de São Teotónio, Viseu, Portugal); Ana Luísa Teixeira (Centro Hospitalar da Cova da Beira, Covilhã, Portugal); António Mendes (Hospital Sousa Martins, Guarda, Portugal); Filomena Pereira (IPO, Lisboa, Portugal); Gustavo Rodrigues (Hospital Lusíadas, Lisboa, Portugal); Alexandra Costa and Ana Teixeira (Centro Hospitalar de Vila Nova de Gaia/Espinho, Vila Nova de Gaia e Espinho, Portugal); Sofia Lima (Hospital Beatriz Ângelo, Loures, Portugal; Hospital de Cascais, Cascais, Portugal; Hospitais Lusíadas, Portugal; Hospitais Luz, Portugal); Érica Laima (Hospital da Luz, Lisboa, Portugal); Maria Ana S. Nunes (Hospital Cruz Vermelha, Lisboa, Portugal); Filipa Prata (Centro Hospitalar Lisboa Norte, Lisboa, Portugal); Pedro Flores

(Hospital CUF Descobertas, Lisboa, Portugal; Laboratórios Germano de Sousa, Portugal); Manuela Brandão (Hospital dos SAMS, Lisboa, Portugal); João Calado Nunes (Hospital Distrital de Santarém, Santarém, Portugal); Rosário Massa (Centro Hospitalar do Médio Tejo, Abrantes, Portugal); Florbela Cunha (Hospital Reynaldo dos Santos, Vila Franca de Xira, Portugal); Paula Correia (Hospital Dr. Fernando da Fonseca, Amadora, Portugal); Anabela Brito (Hospital de Cascais, Cascais, Portugal); João Franco (Hospital Garcia de Orta, Almada, Portugal); Cristina Didelet (Centro Hospitalar do Barreiro Montijo, Barreiro Montijo, Portugal); Estela Veiga (Centro Hospitalar de Setúbal, Setúbal, Portugal); Carla Cruz (Hospital do Espírito Santo, Évora, Portugal); Graça Seves (Centro Hospitalar do Baixo Alentejo, Beja, Portugal); Céu Novais (Hospital Dr. José Maria Grande, Portalegre, Portugal); Maria João Virtuoso and Nancy Guerreiro (Centro Hospitalar do Algarve, Faro e Portimão, Portugal); Amélia Cavaco (Hospital Central do Funchal, Funchal, Portugal); Francisco Gomes (Hospital de Santo Espírito, Angra do Heroísmo, Portugal); Dora Gomes (Hospital da Horta, Horta, Portugal); and Isabel Monteiro (Hospital do Divino Espírito Santo, Ponta Delgada, Portugal).

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
