## [Reviewer comments · Microbiology Spectrum]

Microbiology Spectrum

Continued vaccine breakthrough cases of serotype 3 complicated pneumonia in vaccinated children, Portugal (2016-19)

Catarina silva-costa, Joana Gomes-Silva, Marcos Pinho, Ana Friães, Mario Ramirez, and Melo-Cristino Jose

Corresponding Author(s): Mario Ramirez, Faculdade de Medicina, Universidade de Lisboa

Review Timeline:

Submission Date:	March 28, 2022
Editorial Decision:	May 9, 2022
Revision Received:	June 7, 2022
Accepted:	June 7, 2022

Editor: Eleanor Powell

Reviewer(s): Disclosure of reviewer identity is with reference to reviewer comments included in decision letter(s). The following individuals involved in review of your submission have agreed to reveal their identity: Laura Maria Andrade de Oliveira (Reviewer #2)

Transaction Report:

DOI: <https://doi.org/10.1128/spectrum.01077-22>

May 9, 2022

Dr. Mario Ramirez
Faculdade de Medicina, Universidade de Lisboa
Instituto de Medicina Molecular, Instituto de Microbiologia
Av. Prof. Egas Moniz
Lisboa 1649-028
Portugal

Re: Spectrum01077-22 (Continued vaccine breakthrough cases of serotype 3 complicated pneumonia in vaccinated children, Portugal (2016-19))

Dear Dr. Mario Ramirez:

Thank you for submitting your manuscript to Microbiology Spectrum. After receiving the feedback from reviewers, modifications are necessary before potential publication.

Link Not Available

Sincerely,

Eleanor Powell

Journals Department
Reviewer comments:

Reviewer #1 (Comments for the Author):

I found this study's data pertinent as surveillance studies all around the world are raising concerns about serotype 3 as being one of the most isolated serotype in IPD in children under age of five. This underlines the importance of continuous surveillance of serotype 3 as it was linked to severe cases of IPD. The article gave pertinent data about the vaccination used in Portugal, the methodology of microbiology methods used emphasizing the fact that S.pneumonia is difficult to identify with culture alone and

that other microbiological methods are needed for the identification of PCPP cases especially those linked to serotype 3.

My comments:

Abstract line 23-25: Can you specify the age of the child if they were less than two years or older? And if they were fully vaccinated as in they have received all the recommended doses? It is important to specify these details to be able to talk about vaccine breakthroughs in these patients.

Abstract line 38-40: When talking about reinforcing the continued importance of serotype 3 PCPP, do you mean reinforcing continuous surveillance of serotype 3 PCPP? As serotype 3 is included in PCV13 and it is expected to reduce the exposure to this serotype.

Introduction line 43: *Streptococcus pneumoniae* (pneumococcus) is the leading cause of pneumonia in children, (Can you specify age of children (under 2, under 5, under 16 ?) and the context (where and when)

Introduction line 50: It would be good if you can add that these serotypes are included in the PCV13

Line 51: The PCPP was increasing worldwide or in specific countries? It would be more pertinent for the reader to include these details

Line 59: It would be interesting to include how much time it took for the PCPP in Taiwan to decrease sharply.

For Spain, it would be also pertinent to include information that PCV is included in NIP depending on the region and the regions that included it have seen a decrease in PCPP in children aged X years

Line 75: Can you specify the age of the population where you have seen that 1, 3 and 19A were the dominant causes of PCPP in Portugal ?

Line 91: Can you add the reference of your previous study? So the reader can easily have access to your previous article and learn more about your surveillance system?

For the table 1: It would be good to stratify your yearly results according to the children's age and gender and add a column with fully vaccinated or not while specifying % or number of patients vaccinated with PCV13 or PCV10 or other. It would give the reader a holistic overview of your data and your findings.

Line 197: Can you re-specify the age of your patients?

It would be also good to give results for children under 5 and children under 2. Because this population is the most susceptible to pneumococcal invasive disease.

Discussion

Line 204: Can you add why culture of pleural fluid or blood in PCPP cases is frequently negative and that non-culture methods are essential to identify the etiology of most PCPP cases. (*S. pneumoniae* is hard to identify in culture...)

Line 209-216: It would be interesting to add the age of the patients. Are they all age ? are they children under 18 ?

"we found that almost a third of pIPD cases in 209 Portugal in 2015-2018 were due to PCPP detected using molecular methods" Same for this sentence

"Illustrating this, serotype 3 was the most frequent cause 214 of pIPD in Portugal in 2015-2018 and most cases were detected by molecular methods (n = 215 43/59, 72.8%)"

Line 242: reference is needed

"Despite these encouraging results, if non-culture methods are not used for diagnosis, this could greatly influence the estimates of VE against serotype 3 disease, given the high fraction of this serotype's IPD detected solely by molecular or serologic methods."

Line 250-263: It would be interesting if you could include the age of the population where these findings were found in each country

Line 269: Maybe you could include that serotype 19A had been a dominant serotype before vaccination with PCV-10/13 and is still until today dominant in some PCV-10 contexts in children under the age of five

Line 292: reference is needed

Other references I find interesting to enrich your discussion concerning serotype 3 as I have worked on this in my research project :

1- Jayasinghe S, Liu B, Gidding H, Gibson A, Chiu C, McIntyre P. Long-term Vaccine Impact on Invasive Pneumococcal Disease Among Children With Significant Comorbidities in a Large Australian Birth Cohort. *Pediatr Infect Dis J.* 2019 Sep;38(9):967-973. doi: 10.1097/INF.0000000000002407. PMID: 31408056.

2- Waight PA, Andrews NJ, Ladhani SN, Sheppard CL, Slack MPE, Miller E. Effect of the 13-valent pneumococcal conjugate vaccine on invasive pneumococcal disease in England and Wales 4 years after its introduction: An observational cohort study. *Lancet Infect Dis.* 2015;15(5):535-43.

3- Naucler P, Galanis I, Morfeldt E, Darenberg J, Örtqvist Å, Henriques-Normark B. Comparison of the Impact of Pneumococcal Conjugate Vaccine 10 or Pneumococcal Conjugate Vaccine 13 on Invasive Pneumococcal Disease in Equivalent Populations. *Clin Infect Dis.* 2017;65(11):1780-9.

4- Kandasamy R, Voysey M, Collins S, Berbers G, Robinson H, Noel I, et al. Persistent Circulation of Vaccine Serotypes and Serotype Replacement After 5 Years of Infant Immunization With 13-Valent Pneumococcal Conjugate Vaccine in the United

Kingdom. *J Infect Dis.* 2020 Mar 28;221(8):1361-70.

5- Dagan R, Patterson S, Juergens C, Greenberg D, Givon-Lavi N, Porat N, et al. Comparative immunogenicity and efficacy of 13-valent and 7-valent pneumococcal conjugate vaccines in reducing nasopharyngeal colonization: a randomized double-blind trial. *Clin Infect Dis Off Publ Infect Dis Soc Am.* 2013 Oct;57(7):952-62.

6-Berman-Rosa M, O'Donnell S, Barker M, Quach C. Efficacy and Effectiveness of the PCV-10 and PCV-13 Vaccines Against Invasive Pneumococcal Disease. *Pediatrics.* 2020;145(4).

7-Antachopoulos C, Tsolia MN, Tzanakaki G, Xirogianni A, Dedousi O, Markou G, et al. Parapneumonic pleural effusions caused by *Streptococcus pneumoniae* serotype 3 in children immunized with 13-valent conjugated pneumococcal vaccine. *Pediatr Infect Dis J.* 2014 Jan;33(1):81-3.

8-Madhi F, Godot C, Bidet P, Bahuaud M, Epaud R, Cohen R. Serotype 3 pneumococcal pleural empyema in an immunocompetent child after 13-valent pneumococcal conjugate vaccine. *Pediatr Infect Dis J.* 2014 May;33(5):545-6.
Angoulvant F, Levy C, Grimprel E, Varon E, Lorrot M, Biscardi S, et al. Early impact of 13-valent pneumococcal conjugate vaccine on community-acquired pneumonia in children. *Clin Infect Dis Off Publ Infect Dis Soc Am.* 2014 Apr;58(7):918-24.

9- Godot C, Levy C, Varon E, Picard C, Madhi F, Cohen R. Pneumococcal Meningitis Vaccine Breakthroughs and Failures After Routine 7-Valent and 13-Valent Pneumococcal Conjugate Vaccination in Children in France. *Pediatr Infect Dis J.* 2015 Oct;34(10):e260-263.

10-Moraga-Llop F, Garcia-Garcia J-J, Díaz-Conradi A, Ciruela P, Martínez-Osorio J, González-Peris S, et al. Vaccine Failures in Patients Properly Vaccinated with 13-Valent Pneumococcal Conjugate Vaccine in Catalonia, a Region with Low Vaccination Coverage. *Pediatr Infect Dis J.* 2016 Apr;35(4):460-3.

11-Pírez MC, Algorta G, Chamorro F, Romero C, Varela A, Cedres A, et al. Changes in hospitalizations for pneumonia after universal vaccination with pneumococcal conjugate vaccines 7/13 valent and haemophilus influenzae type b conjugate vaccine in a Pediatric Referral Hospital in Uruguay. *Pediatr Infect Dis J.* 2014 Jul;33(7):753-9.

12-Sings HL, De Wals P, Gessner BD, Isturiz R, Laferriere C, Mclaughlin JM, et al. Effectiveness of 13-Valent Pneumococcal Conjugate Vaccine against Invasive Disease Caused by Serotype 3 in Children: A Systematic Review and Meta-analysis of Observational Studies. *Clin Infect Dis.* 2019;68(12):2135-43.

13-Choi EH, Zhang F, Lu Y-J, Malley R. Capsular Polysaccharide (CPS) Release by Serotype 3 Pneumococcal Strains Reduces the Protective Effect of Anti-Type 3 CPS Antibodies. *Clin Vaccine Immunol CVI.* 2016 Feb;23(2):162-7.

Reviewer #2 (Comments for the Author):

In this study, Silva-Costa et al. conducted a prospective surveillance study to evaluate the effects of introducing PCV13 in the national immunization program on pediatric complicated pneumococcal pneumonia (PCPP) in Portugal. The authors showed the occurrence of vaccine breakthrough cases of PCPP among children, mostly by serotype 3, followed by serotypes 14 and 19A, and shed light on the importance of molecular diagnosis in identifying PCPP. The results of the study reinforce the continued importance of serotype 3 PCPP, even when PCV13 use with almost universal coverage could be expected to reduce exposure to this serotype. This is an interesting and well written research article with clear goals, study parameters and procedures. However, below there are some comments that should be considered by the authors for revision.

- Line 85: the authors should indicate with more detail which are the regions within Portugal the hospital laboratories and pediatric departments included in the study are located.

- Line 91: please indicate the group previous study in the text.

- Line 100: Is there an approval number/registration for the study? If yes, please include this information in the text.

- Lines 104-112: Does the phenotypic identification of colonies suggestive of pneumococci was confirmed by PCR? This is important to test the accuracy of the phenotypic identification and reliability of results, once the occurrence of atypical strains of members of the Mitis group have been reported and can lead to misidentification.

-Line 154: Does the serotype results showed represents the results of capsular reaction test with specific antisera or rPCR? This must be indicated clearer in the text. The authors should indicate with more details how many pneumococcal isolates had their serotype determined by each capsular typing assay (phenotypic and rPCR) performed and the respective percentages of identification.

-Line 162: Please include the addPCV13 in the text (3, 6A, and 19A) to reinforce and make this important information clear to the

reader.

-Line 163: the same above for addPCV10.

-Lines 173-188: please standardize the results' values to include absolute numbers and the respective percentages.

Staff Comments:

Preparing Revision Guidelines

Please return the manuscript within 60 days; if you cannot complete the modification within this time period, please contact me. If you do not wish to modify the manuscript and prefer to submit it to another journal, please notify me of your decision immediately so that the manuscript may be formally withdrawn from consideration by Microbiology Spectrum.

Continued vaccine breakthrough cases of serotype 3 complicated pneumonia in 2 vaccinated children, Portugal (2016-19).

I found this study's data pertinent as surveillance studies all around the world are raising concerns about serotype 3 as being one of the most isolated serotype in IPD in children under age of five. This underlines the importance of continuous surveillance of serotype 3 as it was linked to severe cases of IPD. The article gave pertinent data about the vaccination used in Portugal, the methodology of microbiology methods used emphasizing the fact that S.pneumonia is difficult to identify with culture alone and that other microbiological methods are needed for the identification of PCPP cases especially those linked to serotype 3.

Comments to Authors:

Abstract line 23-25: Can you specify the age of the child if they were less than two years or older? And if they were fully vaccinated as in they have received all the recommended doses? It is important to specify these details to be able to talk about vaccine breakthroughs in these patients.

Abstract line 38-40: When talking about reinforcing the continued importance of serotype 3 PCPP, do you mean reinforcing continuous surveillance of serotype 3 PCPP? As serotype 3 is included in PCV13 and it is expected to reduce the exposure to this serotype.

Introduction line 43: Streptococcus pneumoniae (pneumococcus) is the leading cause of pneumonia in children, (Can you specify age of children (under 2, under 5, under 16 ?) and the context (where and when)

Introduction line 50: It would be good if you can add that these serotypes are included in the PCV13

Line 51: The PCPP was increasing worldwide or in specific countries? It would be more pertinent for the reader to include these details

Line 59: It would be interesting to include how much time it took for the PCPP in Taiwan to decrease sharply.

For Spain, it would be also pertinent to include information that PCV is included in NIP depending on the region and the regions that included it have seen a decrease in PCPP in children aged X years

Line 75: Can you specify the age of the population where you have seen that 1, 3 and 19A were the dominant causes of PCPP in Portugal ?

Line 91: Can you add the reference of your previous study? So the reader can easily have access to your previous article and learn more about your surveillance system?

For the table 1: It would be good to stratify your yearly results according to the children's age and gender and add a column with fully vaccinated or not while specifying % or number of patients vaccinated with PCV13 or PCV10 or other. It would give the reader a holistic overview of your data and your findings.

Line 197: Can you re-specify the age of your patients?

It would be also good to give results for children under 5 and children under 2. Because this population is the most susceptible to pneumococcal invasive disease.

Discussion

Line 204: Can you add why culture of pleural fluid or blood in PCPP cases is frequently negative and that non-culture methods are essential to identify the etiology of most PCPP cases. (*S. pneumoniae* is hard to identify in culture...)

Line 209-216: It would be interesting to add the age of the patients. Are they all age ? are they children under 18 ?

"we found that almost a third of pIPD cases in 209 Portugal in 2015-2018 were due to PCPP detected using molecular methods"

Same for this sentence

"Illustrating this, serotype 3 was the most frequent cause 214 of pIPD in Portugal in 2015-2018 and most cases were detected by molecular methods (n = 215 43/59, 72.8%)"

Line 242: reference is needed

"Despite these encouraging results, if non-culture methods are not used for diagnosis, this could greatly influence the estimates of VE against serotype 3 disease, given the high fraction of this serotype's IPD detected solely by molecular or serologic methods."

Line 250-263: It would be interesting if you could include the age of the population where these findings were found in each country

Line 269: Maybe you could include that serotype 19A had been a dominant serotype before vaccination with PCV-10/13 and is still until today dominant in some PCV-10 contexts in children under the age of five

Line 292: reference is needed

Other references I find interesting to enrich your discussion concerning serotype 3 as I have worked on this in my research project :

1- Jayasinghe S, Liu B, Gidding H, Gibson A, Chiu C, McIntyre P. Long-term Vaccine Impact on Invasive Pneumococcal Disease Among Children With Significant Comorbidities in a Large Australian Birth

Cohort. *Pediatr Infect Dis J*. 2019 Sep;38(9):967-973. doi: 10.1097/INF.0000000000002407. PMID: 31408056.

2- Waight PA, Andrews NJ, Ladhani SN, Sheppard CL, Slack MPE, Miller E. Effect of the 13-valent pneumococcal conjugate vaccine on invasive pneumococcal disease in England and Wales 4 years after its introduction: An observational cohort study. *Lancet Infect Dis*. 2015;15(5):535-43.

3-Naucler P, Galanis I, Morfeldt E, Darenberg J, Örtqvist Å, Henriques-Normark B. Comparison of the Impact of Pneumococcal Conjugate Vaccine 10 or Pneumococcal Conjugate Vaccine 13 on Invasive Pneumococcal Disease in Equivalent Populations. *Clin Infect Dis*. 2017;65(11):1780-9.

4- Kandasamy R, Voysey M, Collins S, Berbers G, Robinson H, Noel I, et al. Persistent Circulation of Vaccine Serotypes and Serotype Replacement After 5 Years of Infant Immunization With 13-Valent Pneumococcal Conjugate Vaccine in the United Kingdom. *J Infect Dis*. 2020 Mar 28;221(8):1361-70.

5- Dagan R, Patterson S, Juergens C, Greenberg D, Givon-Lavi N, Porat N, et al. Comparative immunogenicity and efficacy of 13-valent and 7-valent pneumococcal conjugate vaccines in reducing nasopharyngeal colonization: a randomized double-blind trial. *Clin Infect Dis Off Publ Infect Dis Soc Am*. 2013 Oct;57(7):952-62.

6-Berman-Rosa M, O'Donnell S, Barker M, Quach C. Efficacy and Effectiveness of the PCV-10 and PCV-13 Vaccines Against Invasive Pneumococcal Disease. *Pediatrics*. 2020;145(4).

7-Antachopoulos C, Tsolia MN, Tzanakaki G, Xirogianni A, Dedousi O, Markou G, et al. Parapneumonic pleural effusions caused by *Streptococcus pneumoniae* serotype 3 in children immunized with 13-valent conjugated pneumococcal vaccine. *Pediatr Infect Dis J*. 2014 Jan;33(1):81-3.

8-Madhi F, Godot C, Bidet P, Bahuaud M, Epaud R, Cohen R. Serotype 3 pneumococcal pleural empyema in an immunocompetent child after 13-valent pneumococcal conjugate vaccine. *Pediatr Infect Dis J*. 2014 May;33(5):545-6.

Angoulvant F, Levy C, Grimprel E, Varon E, Lorrot M, Biscardi S, et al. Early impact of 13-valent pneumococcal conjugate vaccine on community-acquired pneumonia in children. *Clin Infect Dis Off Publ Infect Dis Soc Am*. 2014 Apr;58(7):918-24.

9- Godot C, Levy C, Varon E, Picard C, Madhi F, Cohen R. Pneumococcal Meningitis Vaccine Breakthroughs and Failures After Routine 7-Valent and 13-Valent Pneumococcal Conjugate Vaccination in Children in France. *Pediatr Infect Dis J.* 2015 Oct;34(10):e260-263.

10-Moraga-Llop F, Garcia-Garcia J-J, Díaz-Conradi A, Ciruela P, Martínez-Osorio J, González-Peris S, et al. Vaccine Failures in Patients Properly Vaccinated with 13-Valent Pneumococcal Conjugate Vaccine in Catalonia, a Region with Low Vaccination Coverage. *Pediatr Infect Dis J.* 2016 Apr;35(4):460-3.

11-Pírez MC, Algorta G, Chamorro F, Romero C, Varela A, Cedres A, et al. Changes in hospitalizations for pneumonia after universal vaccination with pneumococcal conjugate vaccines 7/13 valent and haemophilus influenzae type b conjugate vaccine in a Pediatric Referral Hospital in Uruguay. *Pediatr Infect Dis J.* 2014 Jul;33(7):753-9.

12-Sings HL, De Wals P, Gessner BD, Isturiz R, Laferriere C, Mclaughlin JM, et al. Effectiveness of 13-Valent Pneumococcal Conjugate Vaccine against Invasive Disease Caused by Serotype 3 in Children: A Systematic Review and Meta-analysis of Observational Studies. *Clin Infect Dis.* 2019;68(12):2135-43.

13-Choi EH, Zhang F, Lu Y-J, Malley R. Capsular Polysaccharide (CPS) Release by Serotype 3 Pneumococcal Strains Reduces the Protective Effect of Anti-Type 3 CPS Antibodies. *Clin Vaccine Immunol CVI.* 2016 Feb;23(2):162-7.

Confidential remarks for the Editors:

Are all the authors' conclusions supported by their data?

The authors in the article came to add valuable data on the persistence of serotype 3 in Portugal despite the vaccination with PCV-13 based on a prospective study. The others based their conclusions on their findings and the results of other similar studies in other countries.

2/ Overall the language used in the article is easy to comprehend.

The article's discussion needs to be further developed with more references as I have included interesting studies in my comment to the authors.

With these minor modifications, the article would be valuable for the scientific community studying serotype 3 breakthroughs and for vaccination committees all around the world.

Reviewer #1 (Comments for the Author):

1) I found this study's data pertinent as surveillance studies all around the world are raising concerns about serotype 3 as being one of the most isolated serotype in IPD in children under age of five. This underlines the importance of continuous surveillance of serotype 3 as it was linked to severe cases of IPD. The article gave pertinent data about the vaccination used in Portugal, the methodology of microbiology methods used emphasizing the fact that *S.pneumonia* is difficult to identify with culture alone and that other microbiological methods are needed for the identification of PCPP cases especially those linked to serotype 3.

We thank the reviewer for the positive appraisal of our work.

2) Abstract line 23-25: Can you specify the age of the child if they were less than two years or older? And if they were fully vaccinated as in they have received all the recommended doses? It is important to specify these details to be able to talk about vaccine breakthroughs in these patients.

We were unable to identify the specific lines the reviewer was referring to since the abstract runs from lines 26-40. However, we would like to clarify that the abstract specifically stated that "Vaccine breakthrough cases were seen among age-appropriately 13-valent PCV vaccinated children". The median age of the children where vaccine breakthroughs were seen was 3-years (range 17-months to 7-years). We have now added a sentence specifically stating this in the text (see answer to point 11), and we have added this information also in the abstract. The relevant sentence, where an error was also corrected, now reads: "Vaccine breakthrough cases were seen among age-appropriately 13-valent PCV vaccinated children (median 3-years, range 17-months to 7-years), mostly with serotype 3 (n=27) but also with serotypes 14 and 19A (n=2 each)."

A detailed description of what was considered age-appropriately vaccinated is given in the text and such a discussion would greatly exceed the word limitations of the abstract so we refrained from going into additional details in the abstract. However, in considering the reviewer's comment, we did notice that we had not specifically stated the vaccinal status of the breakthrough case of the PCV10 vaccinated child. To clarify this, we have added this in the text. The sentence now reads: "One breakthrough was seen with serotype 14 in an age-appropriately 10-valent PCV vaccinated child (...)".

3) Abstract line 38-40: When talking about reinforcing the continued importance of serotype 3 PCPP, do you mean reinforcing continuous surveillance of serotype 3 PCPP? As serotype 3 is included in PCV13 and it is expected to reduce the exposure to this serotype.

Our intended meaning was to highlight the fact that serotype 3 continues to be a major cause of PCPP despite, as the reviewer points out, the fact that serotype 3 being included in PCV13 would be expected to reduce the exposure to this serotype and therefore, potentially, the number of PCPP cases by this serotype. To make this point clearer in the text we have reworded the sentence that now reads: "Our data highlight the importance of molecular diagnostics in identifying PCPP and document the continued importance of serotype 3 PCPP, even when PCV13 use with almost universal coverage could be expected to reduce exposure to this serotype."

4) Introduction line 43: *Streptococcus pneumoniae* (pneumococcus) is the leading cause of pneumonia in children, (Can you specify age of children (under 2, under 5, under 16 ?) and the context (where and when)

The reviewer requests data supporting the statement that “*Streptococcus pneumoniae* (pneumococcus) is the leading cause of pneumonia in children”. As hinted by the reviewer’s questions, the relative importance of the various etiologic agents of CAP was found to be different in studies across different geographies, time and depending on the age of the children. Importantly, the etiology was also found to differ considerably whether the CAP episode required hospitalization or not, and with the extent of microbiology workup performed to identify the causative pathogen. Our intention was to highlight the worldwide importance of *Streptococcus pneumoniae* as a cause of pneumonia. Our statement was based on a study looking at global estimates of the burden of LRTI (GBD 2016 Lower Respiratory Infections Collaborators. 2018. Estimates of the global, regional, and national morbidity, mortality, and aetiologies of lower respiratory infections in 195 countries, 1990–2016: a systematic analysis for the Global Burden of Disease Study 2016. *Lancet Infect Dis* 18:1191–1210) which identified *S. pneumoniae* as the overall leading etiological agent of pneumonia. In the text, data is specifically mentioned for children <5 years and adults >70 years, but in supplemental material (Appendix Figure 5) it is shown the population attributable fractions for all age groups, indicating that pneumococci are responsible for >60% of cases in <18 years (here considered the pediatric population). We believe that such detailed discussion would not add significantly to the paper and we would not like to expand on this in the introduction. To address the reviewer’s concern and to provide better context to the reader we have altered the sentence to read: “*Streptococcus pneumoniae* (pneumococcus) is a leading cause of pneumonia in children worldwide (...)”.

5) Introduction line 50: It would be good if you can add that these serotypes are included in the PCV13

As per the reviewer’s request we now specifically state that these serotypes are included in PCV13. The sentence now reads: “(...) serotypes 1, 3, 7F, 14 and 19A [5], all serotypes included in the 13-valent pneumococcal conjugate vaccine (PCV13).”

6) Line 51: The PCPP was increasing worldwide or in specific countries? It would be more pertinent for the reader to include these details

There are not many studies looking at the temporal trends of PCPP. Where these studies were performed, PCPP was shown to be increasing (see for instance Byington CL, Korgenski K, Daly J, Ampofo K, Pavia A, Mason EO. 2006. Impact of the pneumococcal conjugate vaccine on pneumococcal parapneumonic empyema. *Pediatr Infect Dis J* 25:250–254). We have not identified any study where such trend was contradicted, but it is possible that this was not a universal trend. To address the reviewer’s concern, while not entering into unnecessary detail, we have reworded the sentence that now reads: “Even before the availability of the 7-valent pneumococcal conjugate vaccine (PCV7) the incidence of PCPP was increasing in several countries and this continued despite the use of PCV7 [1,6], which did not include most of the serotypes frequently implicated in PCPP [5].”

7) Line 59: It would be interesting to include how much time it took for the PCPP in Taiwan to decrease sharply.

The decline occurred already in the year the vaccine was introduced nationally in a catch-up program for children 2-5 years (2013). The next year (2014) the national catch-up program was extended to children 1-5 years, and in 2015, a universal vaccination program was instituted nationally for all infants with a 2+1 schedule (2 primary doses at 2 and 4 months of age, and a booster at 12-15 months of age). To address the reviewer's concern we have altered the sentence to read: "(...) but following its introduction in the national immunization program (NIP) with a catch-up program for children 2-5 years-old there was an immediate sharp decrease in PCPP [3]."

8) For Spain, it would be also pertinent to include information that PCV is included in NIP depending on the region and the regions that included it have seen a decrease in PCPP in children aged X years

The reviewer points to the complex vaccination situation in Spain, with different regions having different vaccination strategies. Although a detailed discussion of this situation could be interesting, we do not feel that an expansion of the introduction on this topic would add to the paper. Moreover, the different timeframes of the various studies complicate direct comparisons. Our intention was to point out that different impacts on PCPP have been attributed to the use of PCV13. We would therefore like to leave the text as is.

9) Line 75: Can you specify the age of the population where you have seen that 1, 3 and 19A were the dominant causes of PCPP in Portugal ?

As requested by the reviewer we now indicate that this is referring to individuals seen by pediatric departments (in Portugal <18 years-old), the same age bracket of the current study. The relevant sentence now reads: "(...) we found that serotypes 1, 3 and 19A were the dominant causes of PCPP in children (<18 years-old) (...)"

10) Line 91: Can you add the reference of your previous study? So the reader can easily have access to your previous article and learn more about your surveillance system?

As requested by the reviewer we have introduced the reference to our previous study. The sentence now reads: "There were no changes in the surveillance network from our previous study [4]."

11) For the table 1: It would be good to stratify your yearly results according to the children's age and gender and add a column with fully vaccinated or not while specifying % or number of patients vaccinated with PCV13 or PCV10 or other. It would give the reader a holistic overview of your data and your findings.

The intention of table 1, in which we have found a mistake and corrected it, was to provide an overview of the number of tests performed per year and the number of tests positive for *Streptococcus pneumoniae*. This information was important to demonstrate that there were no significant changes in either during the study period. Information on the age of the patients, requested by the reviewer, is shown in table S1. In the context of increased use of PCV13, particularly among younger children, it would be important to see if there was a temporal change in

age distribution of the cases, a test we did, and which did not support significant changes (presented at the end of the results section). The vaccinal status of the cases, and the serotypes of the pneumococci causing the infection, are presented in table 2. As discussed in the text, there are several types of vaccination status, which we believe cannot be compressed into the two simple categories of “fully vaccinated or not”. As stated in the text, 53% of the cases were in males, indicating an even gender distribution, so additional information on gender distribution does not seem relevant. The yearly serotype distribution of the cases is presented in Table S2, information we used to demonstrate there was no change in the relative number of serotype 3 cases in the study period but that the RR of having a serotype 3 PCPP increased from 2010-15 to 2016-2019. We believe that the table suggested by the reviewer would be confusing for the readers, in trying to display too much information for both PCPP cases as well as for samples negative for *S. pneumoniae*. On the other hand, the information requested is already presented in other tables or in the text, so we do not feel that an additional table is necessary. Still, to address the reviewer’s point and provide more data on the age of the cases where vaccine breakthroughs were seen in PCV13 age-appropriately vaccinated children we have added a sentence reading: “The median age of children age-appropriately vaccinated with PCV13 where a PCV13 serotype was identified was 3-years (range 17-months to 7-years).”

12) Line 197: Can you re-specify the age of your patients? It would be also good to give results for children under 5 and children under 2. Because this population is the most susceptible to pneumococcal invasive disease.

The sentence ending in line 197 (“The same analysis performed on the serotype distribution revealed that for serotype 3 there was no change in 2016-2019 (CA p=0.260) (table S2).”) refers to all individuals <18 years. As indicated in table S1, the number of cases for children <2 years-old was quite low (n=14). This small number of isolates results in a lack of statistical efficiency if we perform analyses stratifying by this age group. On the other hand, the number of isolates from children >5 years-old is quite modest (n=19, 18% of the total), so when we are reporting a global analysis, this is dominated by the group of children ≤5 years-old. So, although we understand the reviewer’s comment that these are the population more at risk for pneumococcal infection, we do not feel that a stratification is warranted. To re-state the age of the patients, as requested by the reviewer, we have changed the sentence that now reads: “The same analysis performed on the serotype distribution of all patients (<18 years) revealed that for serotype 3 there was no change in 2016-2019 (CA p=0.260) (table S2).”

13) Line 204: Can you add why culture of pleural fluid or blood in PCPP cases is frequently negative and that non-culture methods are essential to identify the etiology of most PCPP cases. (*S. pneumoniae* is hard to identify in culture...)

The reviewer raises the interesting point of what could be the reasons why the culture of pleural fluid or blood in PCPP is frequently negative. Although we and many others before us reported that pleural fluid culture is frequently negative (a fact well known of clinical microbiologists), the reasons why this is so remain unknown. This is true even when the patient was not administered antibiotics prior to drainage (which could be a major cause for culture-negative pleural fluid samples). In this context it is important to note that it is also well established that only a fraction (5-20%) of pneumococcal pneumonias are bacteremic, so perhaps it should not be surprising that blood

cultures in PCPP are negative. Moreover, the volume of blood collected in pediatric patients is frequently below the optimal for blood culture, further compounding this issue in pediatrics. Given that a discussion of the causes would be essentially speculative, we would rather not expand this topic in the text.

14) Line 209-216: It would be interesting to add the age of the patients. Are they all age ? are they children under 18 ?

"we found that almost a third of pIPD cases in 209 Portugal in 2015-2018 were due to PCPP detected using molecular methods"

Same for this sentence

"Illustrating this, serotype 3 was the most frequent cause 214 of pIPD in Portugal in 2015-2018 and most cases were detect by molecular methods (n = 215 43/59, 72.8%)"

The points discussed refer to pediatric invasive pneumococcal disease (pIPD) defined in the previous sentence. To clarify that pIPD refers to disease occurring in patients <18 years-old we now specifically state this is the text. The relevant sentence now reads: "This raises the possibility of greatly underestimating the importance of PCPP in overall pediatric (<18 years) invasive pneumococcal disease (pIPD) (...)".

15) Line 242: reference is needed

"Despite these encouraging results, if non-culture methods are not used for diagnosis, this could greatly influence the estimates of VE against serotype 3 disease, given the high fraction of this serotype's IPD detected solely by molecular or serologic methods."

This sentence is not supported by data from the literature but by our own data that indicate we are not detecting a significant fraction of serotype 3 IPD if non-culture methods are not used (see the sentences in the previous question of the reviewer for the relevant data and respective references). We therefore do not feel that including a reference is warranted.

16) Line 250-263: It would be interesting if you could include the age of the population where these findings were found in each country

The reviewer points to the interest of including information on the age of the population where vaccine breakthrough cases were investigated in previous reports of PCV breakthroughs. Like ours, these studies were mostly focused on patients seen in pediatric departments, which age bracket varies with the country considered, but which are similar or a subset of our own age cohort. Also, similarly to our study, the actual age where the vaccine breakthrough cases were seen tended to be in the younger patients (our median age was 3-years). To address the reviewer's concern and to provide the information requested we have changed the relevant sentences that now read: "In the USA, most vaccine breakthrough cases studied in children <5-years occurred in the first year of life and were due to serotypes 19F and 6B with PCV7 vaccination, and serotypes 19A and 3 with PCV13 vaccination [16]. In Ireland, in patients ≤16-years, most PCV13 vaccine breakthroughs occurred in cases of bacteremia without a focus by serotype 19A, although cases with serotypes 3, 6B, 7F, 14 and 19F were also found [17]. In Australia, in empyema cases in patients ≤19-years, most PCV13 vaccine breakthroughs were serotype 3 cases, but there were also cases with serotypes 14, 19A and 19F [1]. In Germany, also only in empyema cases in patients <18-years, most vaccine breakthroughs

were serotype 3 cases, with a minority occurring with serotypes 19A and 1 [9]. In Spain, in patients aged 2-59 months, 24 PCV13 vaccine breakthroughs were detected, mostly associated with complicated pneumonia and serotypes 3 and 19A, although serotypes 1, 6B and 14 were also detected [18].”

17) Line 269: Maybe you could include that serotype 19A had been a dominant serotype before vaccination with PCV-10/13 and is still until today dominant in some PCV-10 contexts in children under the age of five

Serotype 19A was, as the reviewer indicates, an important serotype in IPD, particularly after the introduction in PCV7. The use of PCV13 has led to very significant reductions of 19A IPD and in countries where PCV10 is being used, serotype 19A remains an important cause of IPD, although in some countries decreases were seen in the most recent years. However, we feel that this discussion is tangential to the main message of the paper and would therefore not like to expand on this topic in the text.

18) Line 292: reference is needed

In most countries a 2+1 schedule is used, whereas in the United States a 3+1 schedule is used. In Portugal, we went from a 3+1 schedule before use in the NIP to a 2+1 schedule. This sentence is supported by observations of vaccine failures in countries with either of these vaccination schedules. As requested by the reviewer we now specifically indicate the relevant references at the end of the sentence.

19) Other references I find interesting to enrich your discussion concerning serotype 3 as I have worked on this in my research project :

We thank the reviewer for the list of references provided. Although they offer interesting insights into several aspects of vaccine efficacy and serotype 3 biology, we would like to keep the focus of the paper and have therefore not expanded the introduction nor the discussion and have not added any of them at this time.

Reviewer #2

1) In this study, Silva-Costa et al. conducted a prospective surveillance study to evaluate the effects of introducing PCV13 in the national immunization program on pediatric complicated pneumococcal pneumonia (PCPP) in Portugal. The authors showed the occurrence of vaccine breakthrough cases of PCPP among children, mostly by serotype 3, followed by serotypes 14 and 19A, and shed light on the importance of molecular diagnosis in identifying PCPP. The results of the study reinforce the continued importance of serotype 3 PCPP, even when PCV13 use with almost universal coverage could be expected to reduce exposure to this serotype. This is an interesting and well written research article with clear goals, study parameters and procedures. However, below there are some comments that should be considered by the authors for revision.

We thank the reviewer for the positive appraisal of our work.

2) Line 85: the authors should indicate with more detail which are the regions within Portugal the hospital laboratories and pediatric departments included in the study are located.

Our surveillance network includes hospitals in all regions of Portugal, including the islands. Since the network includes most tertiary hospitals in all regions (units where drainage of pleural fluid is most likely to occur), we believe that we cover the entire Portuguese population. To address the reviewer's concern and to better convey this to the reader we have altered the sentence that now reads: "(...) pleural fluid samples of patients <18 years from 61 hospital laboratories and pediatric departments in all regions of Portugal."

3) Line 91: please indicate the group previous study in the text.

As also requested by reviewer #1 we have introduced the reference to our previous study. The sentence now reads: "There were no changes in the surveillance network from our previous study [4]."

4) Line 100: Is there an approval number/registration for the study? If yes, please include this information in the text.

We thank the reviewer for pointing to the missing information, which we have now added to the text. The relevant sentence now reads: "This study was approved by the Institutional Review Board of the Centro Académico de Medicina de Lisboa (246/17)".

5) Lines 104-112: Does the phenotypic identification of colonies suggestive of pneumococci was confirmed by PCR? This is important to test the accuracy of the phenotypic identification and reliability of results, once the occurrence of atypical strains of members of the Mitis group have been reported and can lead to misidentification.

The reviewer points to the possibility of misidentification of *Streptococcus pneumoniae*. However, we consider this extremely unlikely. We have used a set of phenotypic methods which, together, constitutes the gold-standard for the identification of pneumococci, namely: colony morphology and hemolysis on blood agar plates, optochin susceptibility and bile solubility. Moreover, in all our isolates a serotype could be identified (there were no non-typable isolates, see next point). Taken together this data give us a high confidence on the species identification of the isolates.

6) Line 154: Does the serotype results showed represents the results of capsular reaction test with specific antisera or rPCR? This must be indicated clearer in the text. The authors should indicate with more details how many pneumococcal isolates had their serotype determined by each capsular typing assay (phenotypic and rPCR) performed and the respective percentages of identification.

We thank the reviewer for raising this point which allows us to clarify an important issue which was unclear in the text. As stated in the answer to the previous point and stated in the methods section, all isolates had a serotype identified by serology. All the samples where a serotype was not identified were patient samples where serotype identification was attempted by rPCR. As we state in the

following text to the line indicated by the reviewer, we believe this to be most probably because the serotypes in question are not present in the rPCR schema we used. To address the reviewer's concern and to clarify this point in the text we have altered the text to read: "We were able to serologically identify the serotype in all (n=11) isolates available and by rPCR in 86.2% (n=75) of the samples where *S. pneumoniae* was detected by rPCR (table 2). In the remaining 12 pleural fluid samples, most likely the serotype of the causative pneumococcus was not included in those covered by the rPCR schema used and these samples were classified as "not identified"."

7) Line 162: Please include the addPCV13 in the text (3, 6A, and 19A) to reinforce and make this important information clear to the reader.

As requested by the reviewer we now re-state in this sentence that the addPCV13 serotypes are 3, 6A, and 19A. The relevant sentence now reads: "Overall, the addPCV13 serotypes (3, 6A and 19A) were responsible for most infections (n=69, 70.4%)."

8) Line 163: the same above for addPCV10.

As requested by the reviewer we now re-state in this sentence that the addPCV10 serotypes are 1, 5, and 7F. The relevant sentence now reads: "PCV7 serotypes accounted for 6.1% of the cases (n=6) and the addPCV10 serotypes (1, 5, 7F) accounted for an even smaller proportion of the cases (n=2, 2%)."

9) Lines 173-188: please standardize the results' values to include absolute numbers and the respective percentages.

As requested by the reviewer we now indicate the percentages and the absolute numbers of all the novel data discussed. The relevant sentences now read: "PCV7 serotypes accounted for 6.1% of the cases (n=6) and the addPCV10 serotypes (1, 5, 7F) accounted for an even smaller proportion of the cases (n=2, 2%). In 21 cases (21.4%), the infection was caused by an NVT, a higher proportion than in 2010-2015 (11%) [4], but the difference was not statistically supported (p=0.056)."

June 7, 2022

Dr. Mario Ramirez
Faculdade de Medicina, Universidade de Lisboa
Instituto de Medicina Molecular, Instituto de Microbiologia
Av. Prof. Egas Moniz
Lisboa 1649-028
Portugal

Re: Spectrum01077-22R1 (Continued vaccine breakthrough cases of serotype 3 complicated pneumonia in vaccinated children, Portugal (2016-19))

Dear Dr. Mario Ramirez:

I'm pleased to inform you that your manuscript has been accepted, and I am forwarding it to the ASM Journals Department for publication. You will be notified when your proofs are ready to be viewed.

Sincerely,

Eleanor Powell
Editor, Microbiology Spectrum
